# Serum bridging molecules drive candidal invasion of human but not mouse endothelial cells

Quynh T. Phan[1], Norma V. Solis[1], Jianfeng Lin[1], Marc Swidergall[1,2], Shakti Singh[1], Hong Liu[1], Donald C. Sheppard[3], Ashraf S. Ibrahim[1,2], Aaron P. Mitchell[4], Scott G. Filler [1,2] *

1 Institute for Infection and Immunity, Division of Infectious Diseases, The Lundquist Institute for Biomedical Innovation at Harbor-UCLA Medical Center, Torrance, California, United States of America, 2 David Geffen School of Medicine at UCLA, Los Angeles, California, United States of America, 3 Department of Microbiology and Immunology, Faculty of Medicine, McGill University, Montreal, Quebec, Canada, 4 Department of Microbiology, University of Georgia, Athens, Georgia, United States of America

* sfiller@ucla.edu

**Data Availability Statement:** All relevant data are within the manuscript and its Supporting Information files.

## Abstract

During hematogenously disseminated candidiasis, blood borne fungi must invade the endothelial cells that line the blood vessels to infect the deep tissues. Although *Candida albicans*, which forms hyphae, readily invades endothelial cells, other medically important species of *Candida* are poorly invasive in standard in vitro assays and have low virulence in immunocompetent mouse models of disseminated infection. Here, we show that *Candida glabrata*, *Candida tropicalis*, *Candida parapsilosis*, and *Candida krusei* can bind to vitronectin and high molecular weight kininogen present in human serum. Acting as bridging molecules, vitronectin and kininogen bind to αv integrins and the globular C1q receptor (gC1qR), inducing human endothelial cells to endocytose the fungus. This mechanism of endothelial cell invasion is poorly supported by mouse endothelial cells but can be restored when mouse endothelial cells are engineered to express human gC1qR or αv integrin. Overall, these data indicate that bridging molecule-mediated endocytosis is a common pathogenic strategy used by many medically important *Candida spp.* to invade human vascular endothelial cells.

## Author summary

The invasion of vascular endothelial cells is a key step in the pathogenesis of hematogenously disseminated candidiasis. How species of *Candida* other than *C. albicans* invade endothelial cells is poorly understood. Here, we demonstrate that *Candida glabrata* and other *Candida* spp. adhere to and invade human endothelial cells by utilizing the serum proteins kininogen and vitronectin as bridging molecules between the fungus and the host cell. When bound to the surface of the fungi, these serum proteins interact with the globular C1q receptor (gC1qR) and αv integrins on the endothelial cell surface, inducing endocytosis. This process occurs with human but not mouse endothelial cells but can be restored in mouse endothelial cells that express human gC1qR or integrin αv. Thus,

**Funding:** This work was supported in part by NIH grant R01AI124566 to S.G.F., R00DE026856 to M. S., and R01AI141202 to A.S.I. The collection of umbilical cords for this project was supported in part by the National Center for Advancing Translational Sciences through UCLA CTSI Grant UL1TR001881. The funders played no role in the study design, data collection and analysis, decision to publish, or preparation of the manuscript.

bridging molecule-mediated endocytosis is a common mechanism by which medically important *Candida* spp. invade human vascular endothelial cells.

## Introduction

Despite the widespread use of antifungal agents, disseminated candidiasis continues to be a serious problem in hospitalized patients. Previously, *Candida albicans* was the most common cause of candidemia [1]. However, the epidemiology of this disease has changed, and *C. albicans* now accounts for less than half of cases of candidemia. In fact, the combined incidence of infections caused by *Candida glabrata*, *Candida parapsilosis*, and *Candida tropicalis* now exceeds the incidence of infections caused by *C. albicans* [2,3]. Even though the causative agents of candidemia have changed, this infection remains highly lethal; approximately 40% of patients with candidemia die, even with currently available therapy [2,3]. A deeper understanding of the pathogenesis of this disease is essential for developing new strategies to prevent and treat invasive candidal infections.

During hematogenously disseminated candidiasis, blood-borne organisms must invade the endothelial cell lining of the vasculature to reach the target organs [4]. Because escape from the vasculature is a key step in the pathogenesis of the infection, there is intense interest in dissecting the mechanism by which this process occurs. It is known that *C. albicans* invades endothelial cells by forming filamentous hyphae that express invasins such as Als3 and Ssa1. These invasins interact with specific host cell receptors including N-cadherin and GP96, thereby stimulating endothelial cells to endocytose the fungus [5–10].

While many of the factors that enable *C. albicans* to invade endothelial cells have been delineated, the mechanisms by which other medically important species of *Candida* invade endothelial cells remains poorly understood. A major obstacle to understanding this process is that organisms such as *C. glabrata* and *C. tropicalis* that do not form true hyphae on endothelial cells have greatly impaired capacity to invade these cells in standard in vitro assays [11,12]. These organisms also have limited virulence in immunocompetent mice [13–15]. By contrast, these organisms are still able to cross the endothelial cell lining of the vasculature in humans and infect target organs during disseminated infection [2,3]. Indeed, patients with candidemia caused by *C. glabrata*, which grows only in the yeast form in vivo, have at least as high mortality as those with candidemia due to *C. albicans* [2,16]. While some of the mortality associated with *C. glabrata* fungemia may be attributed to antifungal resistance and patient co-morbidities, it is clear that afilamentous *C. glabrata* is highly virulent in humans.

These data suggest that yeast-phase *Candida* spp. must be able to penetrate endothelial cells in vivo by a mechanism that is not evident in standard in vitro invasion assays. Most assays of *Candida* invasion are performed using media that contain either heat-inactivated serum or no serum at all. Here we demonstrate that when yeast-phase *Candida* spp. such as *C. glabrata* are incubated with either fresh human serum or plasma, two proteins, high molecular weight kininogen and vitronectin bind to the fungal surface. Acting as bridging molecules, these serum proteins interact with the globular C1q receptor (gC1qR; also known as p33/HABP) and αv integrins on the surface of human endothelial cells and induce adherence and endocytosis of the organisms. When *C. glabrata* is coated with either human or mouse serum, there is minimal endocytosis by mouse endothelial cells, suggesting a key limitation of the mouse model to study vascular invasion by yeast phase *Candida* spp. This mouse-specific defect in endocytosis can be rescued in vitro by expressing either human gC1qR or human αv integrin in mouse

endothelial cells. Thus, we delineate a previously unexplored mechanism by which fungi can invade human endothelial cells.

# Results

## Serum and plasma enhance the endocytosis of *Candida spp*

Previously, we found that yeast-phase *C. albicans*, such as live *efg1Δ/Δ cph1Δ/Δ* mutant cells or killed wild-type yeast, are very poorly endocytosed by human endothelial cells in vitro [12]. A limitation of these previous experiments is that they were performed in serum-free media. It is known that serum proteins can act as bridging molecules and mediate the adherence of bacteria to endothelial cells [17,18]. Also, although yeast-phase *C. parapsilosis* cells are poorly endocytosed by endothelial cells in the absence of serum in vitro, this process is much more efficient in the presence of serum [19]. Therefore, we investigated whether serum components could act as bridging molecules between *Candida* spp. and endothelial cells. Live *C. glabrata* yeast and methanol killed, yeast phase *C. albicans* cells were incubated in 20% pooled human serum that was either fresh or heat-inactivated. Killed *C. albicans* cells were used in these experiments because live organisms germinate when exposed to serum [20]. The fungal cells were then rinsed and then added to human umbilical vein endothelial cells. When the organisms were incubated with heat-inactivated serum, few cells were endocytosed, similarly to control organisms that had been incubated in serum-free medium (Fig 1A and 1B). When the organisms were incubated in fresh serum, the number of endocytosed cells increased by 8- to 9-fold. Incubating *C. glabrata* and *C. albicans* with fresh serum also increased the number of cell-associated organisms, a measure of adherence (Fig 1C and 1D). To verify that serum could enhance the endothelial cell interactions of live *C. albicans*, we tested an *efg1Δ/Δ cph1Δ/Δ* mutant strain that remains in the yeast phase when exposed to serum [15]. The endocytosis and adherence of this strain were increased when it was incubated in fresh serum as compared to heat-inactivated serum (S1A and S1B Fig). Although fresh serum significantly enhanced the endothelial cell endocytosis of three additional *C. glabrata* blood isolates, it only increased the adherence of one of these strains (S1C and S1D Fig). Fresh serum also improved the endocytosis and adherence of live, yeast-phase *C. parapsilosis*, *Candida krusei*, and *Candida auris*, but not *C. tropicalis* (Fig 1E and 1F). Fresh human plasma was at least as effective as fresh human serum at enhancing the endocytosis and adherence of *C. glabrata* (S1E and S1F Fig), indicating that both plasma and serum contain factors that strongly enhance the endothelial cell interactions of yeast-phase organisms.

Although fresh serum significantly enhanced the endocytosis and adherence of multiple species of *Candida*, it only increased the adherence of *S. cerevisiae* but had no effect on endocytosis (S1G and S1H Fig). Thus, the bridging molecules that bind to *Candida* spp. appear to be non-functional after they bind to *S. cerevisiae*.

To verify that the serum-coated organisms were being endocytosed, endothelial cells were infected with serum-coated *C. glabrata*, fixed and then stained for actin. We observed that actin microfilaments coalesced around *C. glabrata* cells, a hallmark of endocytosis (Fig 1G). When endothelial cells were treated with cytochalasin D to depolymerize actin, the endocytosis of serum-coated organisms was significantly decreased (Fig 1H). Cytochalasin D also reduced the number of adherent organisms (Fig 1I). Collectively, these data suggest the model that heat-labile serum proteins function as bridging molecules that induce endothelial cells to endocytose yeast-phase *Candida* spp.

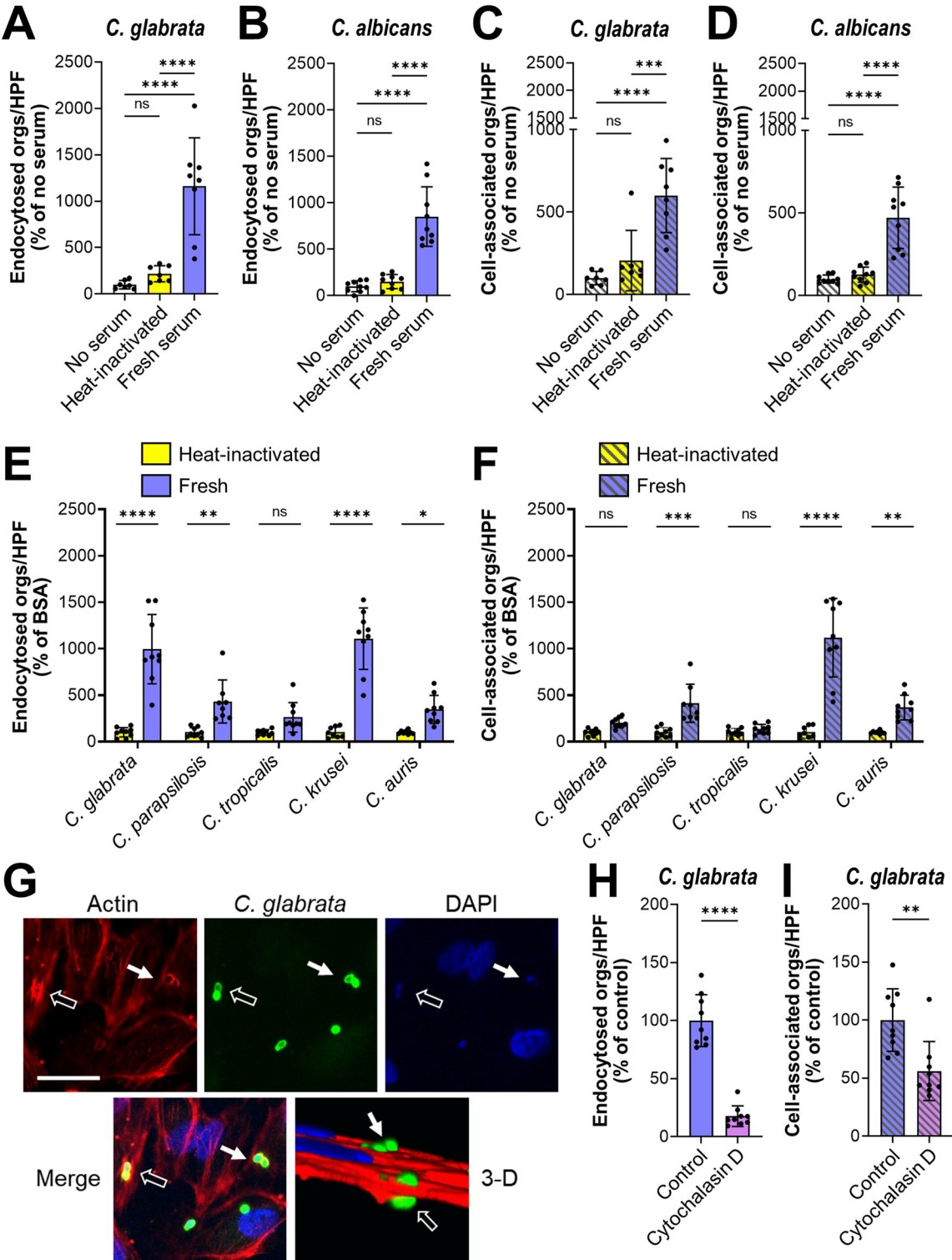

**Fig 1. Serum coating increases the endocytosis of *Candida spp.* by human endothelial cells.** (A-D), Effects of fresh and heat-inactivated human serum on the number of organisms that were endocytosed by and cell-associated (a measure of adherence) with human umbilical vein endothelial cells. (A) Endocytosis of live *C. glabrata*, (B) endocytosis of killed wild-type *C. albicans* yeast, (C) cell-association of live *C. glabrata*, (D) cell-association of killed wild-type *C. albicans* yeast. (E-F) endocytosis (E) and cell-association (F) of live cells of the indicated *Candida spp.* (G) Confocal micrographs showing the accumulation of phalloidin-stained endothelial cell actin

around endocytosed *C. glabrata* cells. Results are representative of 3 independent experiments. Arrows indicate the *C. glabrata* cells and the actin that has accumulated around them. Scale bar, 10 μm. (H and I) Effects of cytochalasin D on the endocytosis (H) and cell-association (I) of live *C. glabrata* coated with fresh human serum. Results are the mean ± SD of 3 experiments each performed in triplicate. Orgs/HPF, organisms per high-power field; ns, not significant; $^*P < 0.05$, $^{**}P < 0.01$, $^{***}P < 0.001$, $^{****}P < 0.0001$ by ANOVA with the Dunnett's test for multiple comparisons (A-F) or the Student's t-test (H and I).

## The globular C1q receptor (gC1qR) and αv integrins are endothelial cell receptors for serum-coated yeast-phase *C. glabrata*

To identify potential endothelial cell receptors for serum-coated organisms, we used a knowledge-directed approach. We selected gC1qR for analysis because it is known to bind to several different serum proteins [21]. To determine if gC1qR interacted with serum-coated *C. glabrata* cells, we incubated endothelial cell membrane proteins with *C. glabrata* coated with either fresh or heat-inactivated serum. After extensively rinsing the cells to remove the unbound proteins, the remaining bound proteins were eluted with 6M urea and separated by SDS-PAGE. By immunoblotting with an anti-gC1qR monoclonal antibody, we determined that more gC1qR was associated with *C. glabrata* cells coated with fresh serum relative to cells coated with heat-inactivated serum (Fig 2A). To determine the functional significance of this interaction, we used siRNA to knockdown gC1qR. We found that gC1qR siRNA significantly inhibited the endocytosis of serum-coated *C. glabrata* (Figs 2B and S2). The gC1qR siRNA also slightly inhibited *C. glabrata* adherence (Fig 2C). Because gC1qR is known to be expressed both intracellularly and on the cell surface [22,23], siRNA knockdown likely depleted both pools of this protein. To verify that surface-expressed gC1qR was required for the endocytosis of serum-coated *C. glabrata*, we tested two different anti-gC1qR monoclonal antibodies. Antibody 74.5.2, which recognizes the high molecular weight kininogen binding site in the C-terminus of the gC1qR [24,25], reduced endocytosis by 45% but did not significantly affect adherence (Fig 2D and 2E). By contrast, antibody 60.11, which is directed against the C1q binding site in the N-terminus of the gC1qR, had no effect on either endocytosis or adherence. Collectively, these data suggest that the gC1qR functions as an endothelial cell receptor for serum-coated *C. glabrata*. They also indicate that the C-terminus of gC1qR, which contains the binding site for high molecular weight kininogen, plays a key role in bridging molecule-mediated endocytosis.

The finding that blocking gC1qR resulted in incomplete inhibition of endocytosis prompted us to search for additional endothelial cell receptors for serum-coated *C. glabrata*. Because integrins bind to serum proteins that could potentially act as bridging molecules, we tested whether siRNA knockdown of either integrin αv or α5 could reduce the interactions of serum-coated *C. glabrata* with human endothelial cells. We found that knockdown of integrin αv, but not integrin α5 significantly reduced the endocytosis and adherence of *C. glabrata* (Figs 3A and 3B and S3A). Integrin αv can form a heterodimer with integrin β3 and β5. We therefore tested the effects of antibodies against integrins αvβ3 and αvβ5 and determined that both antibodies inhibited endocytosis and adherence by approximately 45% (Fig 3C–3F). Although some antibodies against integrins αvβ3 and αvβ5 also bind to the surface of *C. albicans* [26,27], flow cytometry confirmed that the monoclonal antibodies used in our experiments did not bind to *C. glabrata* (S3B Fig).

Notably, blocking gC1qR and integrins αvβ3 and αvβ5 simultaneously decreased endocytosis to the basal level induced by heat-inactivated serum, but did not further decrease the adherence of *C. glabrata* (Figs 3G and 3H and S4A and S4B). Collectively, these results indicate that the gC1qR and integrins αvβ3 and αvβ5 are key receptors that mediate the endocytosis of serum coated organisms.

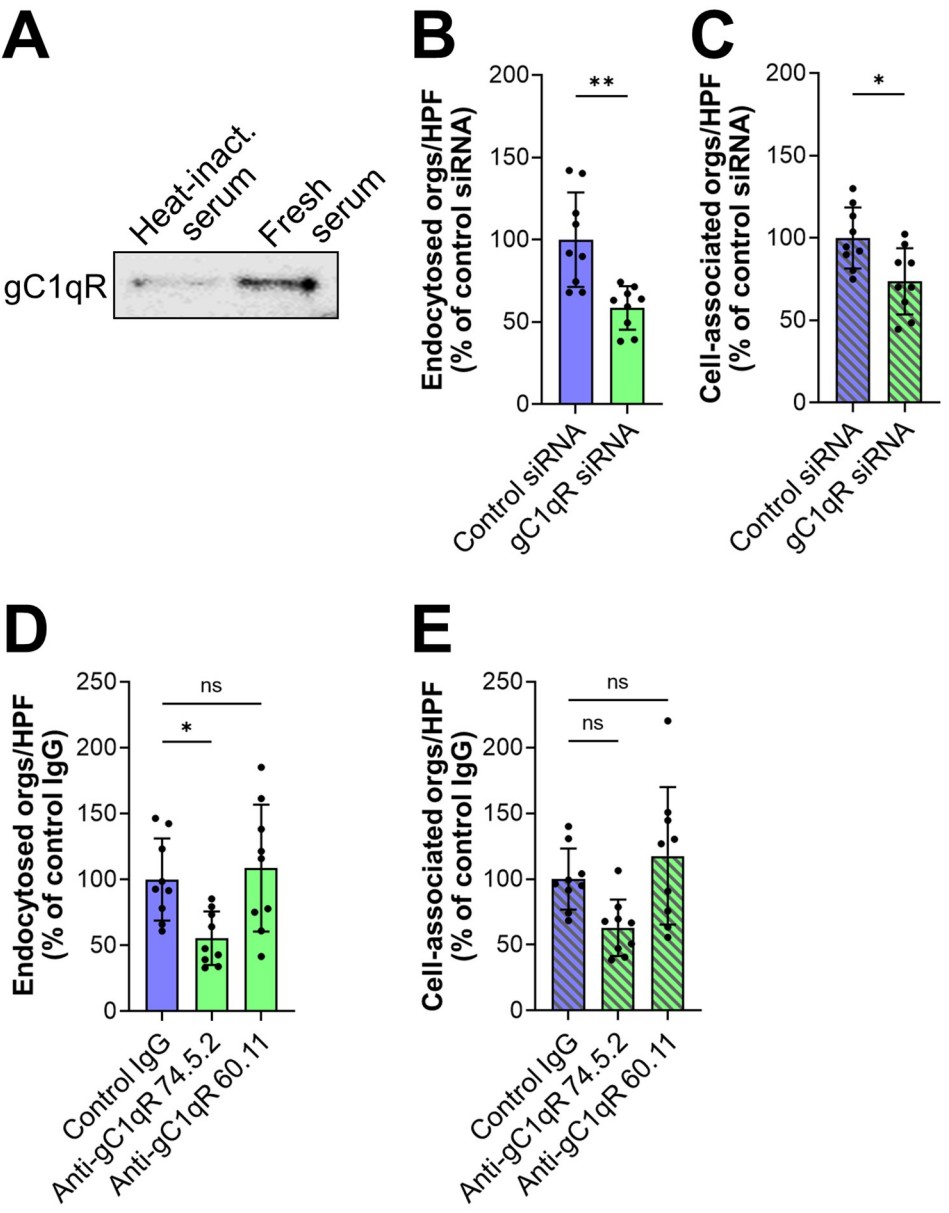

**Fig 2. The globular C1q receptor (gC1qR) is a human endothelial cell receptor for serum-coated _C. glabrata_.** (A) Western blot showing that endothelial cell gC1qR binds to _C. glabrata_ coated with fresh serum. Results are representative of 3 independent experiments. (B-E) Effects of inhibiting gC1qR function with siRNA knockdown (B and C) and specific monoclonal antibodies (D and E) on the endocytosis (B and D) and cell-association (C and E) of _C. glabrata_ coated with fresh serum. Antibody 74.5.2 recognizes the high molecular weight kininogen binding site in the C-terminus of the gC1qR and antibody 60.11 is directed against the C1q binding site in the N-terminus of the gC1qR. Results shown in (B-E) are the mean ± SD of 3 experiments each performed in triplicate. Heat inact., heat inactivated serum; orgs/HPF, organisms per high-power field; ns, not significant, *$P < 0.05$, **$P < 0.01$ by Student's t-test (B and C) or ANOVA with the Dunnett's test for multiple comparisons (D and E).

To further explore the relationship among the gC1qR and the integrins αvβ3 and αvβ5, we infected endothelial cells with serum-coated _C. glabrata_, stained them with antibodies against the three receptors, and then imaged them with confocal microscopy. We observed that all three receptors accumulated around organisms coated with fresh serum (Fig 4A), whereas

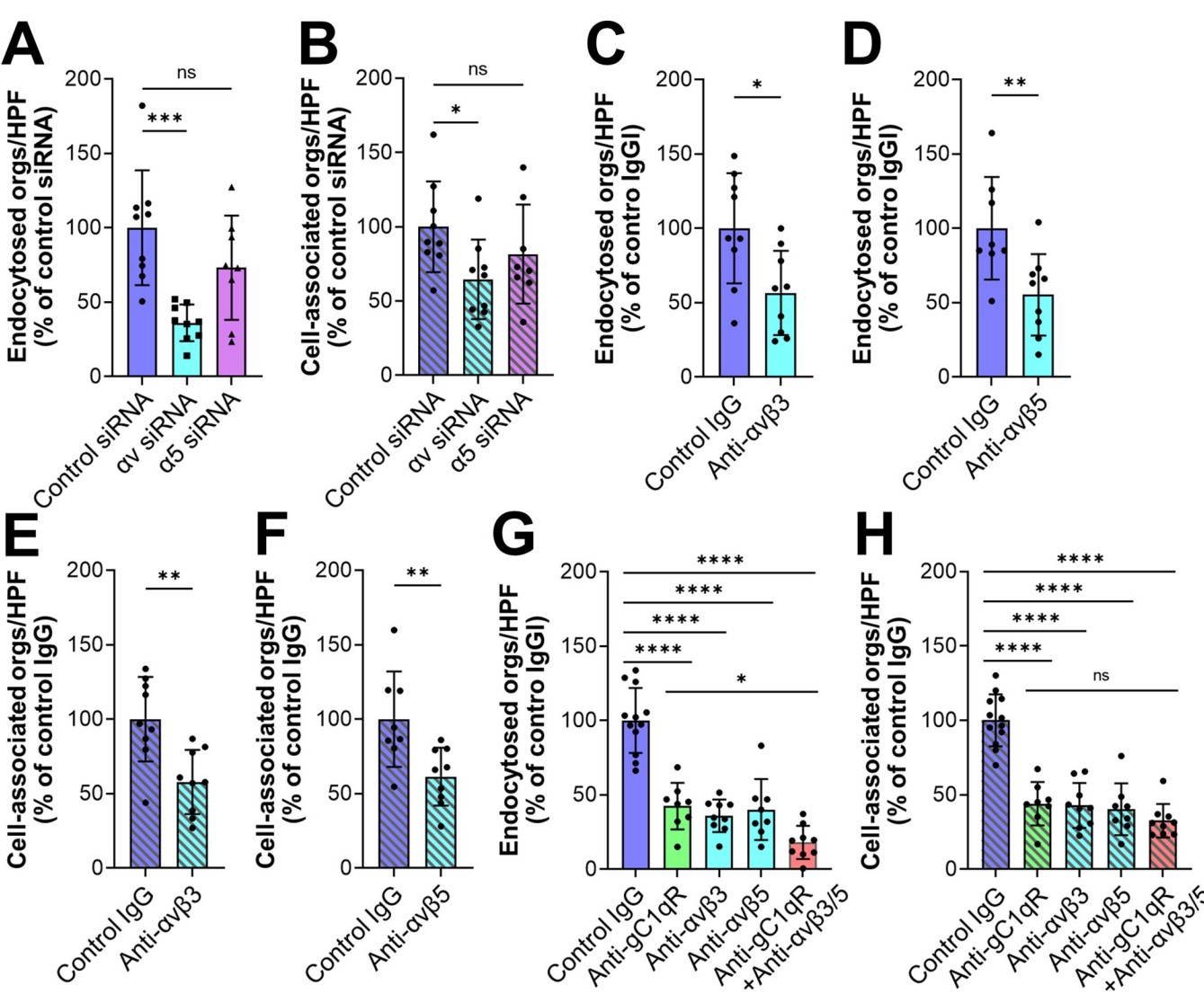

**Fig 3. Integrins αvβ3 and αvβ5 are human endothelial cell receptors for serum-coated *C. glabrata*.** (A-F) Effects of inhibiting αv integrin function with siRNA knockdown (A and B or)specific monoclonal antibodies (C-F) and on the endocytosis (A, C, D) and cell-association (B, E, F) of serum-coated *C. glabrata*. (G and H) Inhibition of gC1qR (with monoclonal antibody 74.5.2) and αv integrins has an additive effect on decreasing the endocytosis (G) but not cell-association of serum-coated *C. glabrata* (H). Results are the mean ± SD of 3 experiments, each performed in triplicate. Orgs/HPF, organisms per high power field; ns, not significant; *$P < 0.05$, **$P < 0.01$, ***$P < 0.001$, ****$P < 0.0001$ by ANOVA with the Dunnett's test for multiple comparisons (A, B, G, H) or the Student's t-test (C-F).

there was minimal accumulation of these receptor around organisms coated with heat-inactivated serum (Fig 4B). Collectively, these results support the model that when serum proteins bind to a *C. glabrata* cell, they interact with gC1qR and integrins αvβ3 and αvβ5 on the endothelial cell surface, which causes organism to adhere to and be endocytosed by the endothelial cell.

### High molecular weight kininogen and vitronectin are bridging molecules that mediate the endocytosis of serum-coated organisms

Next, we sought to identify potential bridging molecules that mediate the binding of serum-coated organisms to gC1qR and integrins αvβ3 and αvβ5. After incubating *C. glabrata* cells in

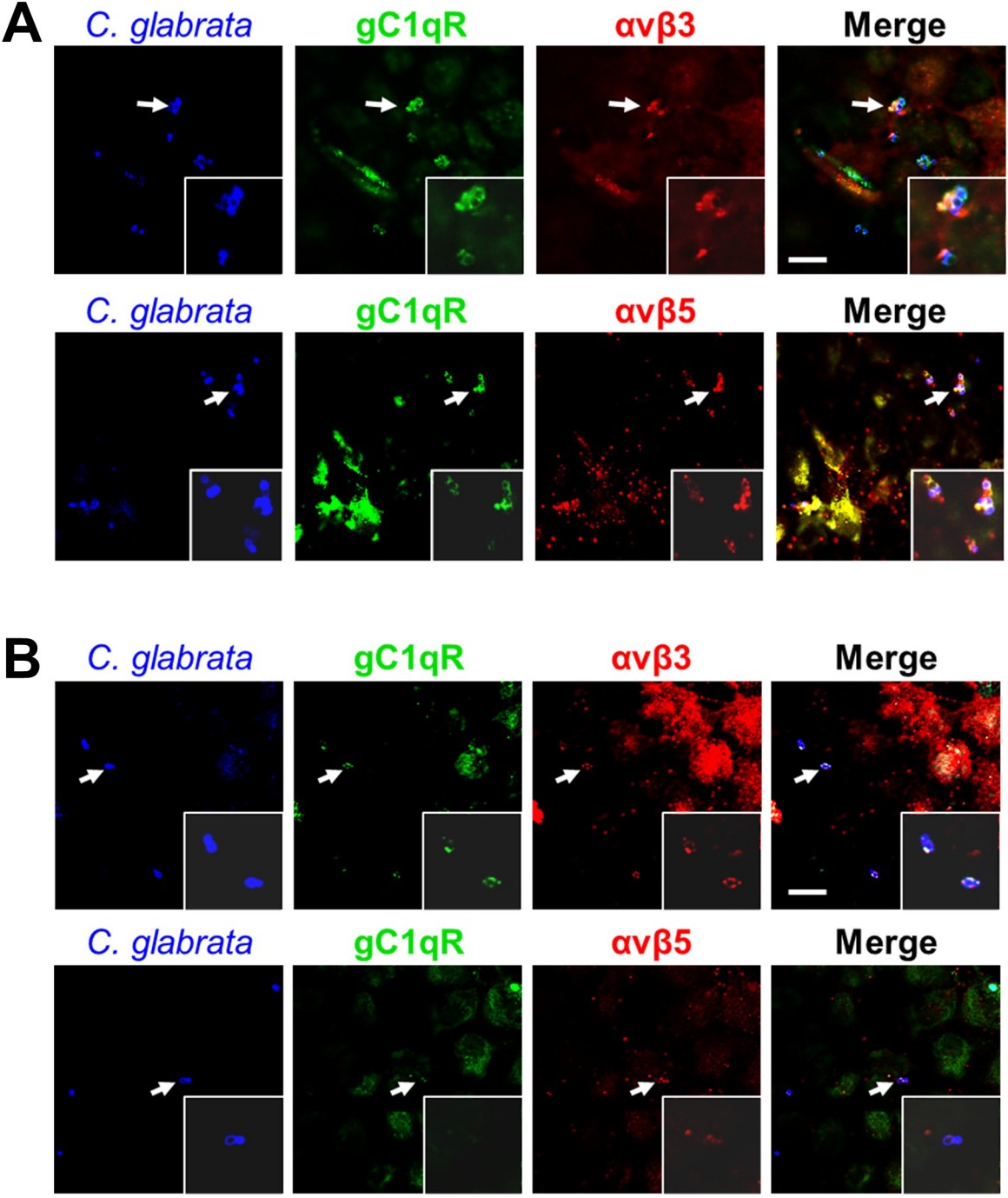

**Fig 4. Integrins αvβ3 αvβ5 are accumulate around *C. glabrata* cells coated with fresh human serum.** Confocal micrographs of human endothelial cells showing the accumulation of gC1qR and integrins αvβ3 and αvβ5 around *C. glabrata* cells coated with either fresh (A) or heat-inactivated (B) serum. Arrows show the organisms that are in the magnified insets. Representative results of 3 independent experiments. Scale bar, 10 μm.

fresh serum, we rinsed them extensively and eluted the bound serum proteins with HCl followed by Tris neutralization. The eluted proteins were separated by SDS-PAGE and analyzed by Western blotting to detect proteins that are known to bind to these receptors. Two proteins,

high molecular weight kininogen and vitronectin, were identified. These proteins could be eluted from *C. glabrata* when the cells were incubated in fresh serum, but not heat-inactivated serum (Fig 5A). The binding of these proteins to *C. glabrata* was functionally significant because antibodies against kininogen and vitronectin significantly inhibited endocytosis, but had no effect on adherence (Figs 5B and S5A). Also, coating *C. glabrata* cells with kininogen and vitronectin significantly enhanced their endocytosis by and adherence to endothelial cells relative to organisms coated with BSA (Figs 5C and S5B). This augmentation progressively increased with the multiplicity of infection. Collectively, the results indicate that kininogen and vitronectin function as bridging molecules that induce endothelial cells to endocytose *C. glabrata*.

High molecular weight kininogen is cleaved by kallikrein and other proteases, releasing bradykinin from the larger protein. The remaining protein, called HKa, consists of a 62 kDa heavy chain that is linked by a disulfide bond to a 56 kDa light chain [28]. By immunoblotting with specific monoclonal antibodies and looking for bands of the appropriate molecular mass, we found that both the heavy and light chains of HKa bound to *C. glabrata* (Fig 5A). Both of these chains were bound by *C. glabrata* when the cells were incubated in fresh human serum, but not with heat-inactivated serum. Collectively, these results suggest that high molecular weight kininogen is cleaved to HKa, which then binds to *C. glabrata*.

Using flow cytometry, we analyzed the relationship between the binding of kininogen and vitronectin to *C. glabrata*. We found that when the organisms were incubated with kininogen alone, very little protein bound to them (Fig 5D and 5E). When the organisms were incubated with kininogen in the presence of vitronectin, kininogen binding increased significantly. By contrast, vitronectin bound to the organisms both in the presence and absence of kininogen (Fig 5D and 5F). These results suggest the model that vitronectin binds to the organism and facilitates the binding of kininogen.

To determine if kininogen and vitronectin could function as bridging molecules by themselves, we incubated *C. glabrata* cells with these proteins, either alone or in combination, and then measured their endocytosis by endothelial cells. When the organisms were incubated with kininogen alone, few organisms were endocytosed, similarly to control organisms that had been incubated in BSA (Fig 5G). When the organisms were incubated in vitronectin alone, endocytosis increased significantly, and it increased even more when the organisms were incubated in both kininogen and vitronectin. The combination of kininogen and vitronectin also significantly increased the adherence of the organisms, while kininogen and vitronectin alone had no effect (S5C Fig). Collectively, these data indicate that the human serum proteins kininogen and vitronectin function as bridging molecules that enhance adherence and induce endocytosis of *C. glabrata* by human endothelial cells.

We next investigated whether kininogen and vitronectin could act as bridging molecules for *C. albicans*. Using flow cytometry, we verified that these proteins bound to methanol killed *C. albicans* yeast (S5D–S5F Fig). We also analyzed the endothelial cell interactions of live *C. albicans* cells that had been incubated in kininogen and vitronectin prior to being added to these host cells. These experiments were feasible because, unlike serum, kininogen and vitronectin did not induce significant filamentation. We found that preincubating organisms with these proteins significantly enhanced the endocytosis and adherence of the *C. albicans efg1Δ/Δ cph1Δ/Δ* mutant, which remained in the yeast phase while in contact with the endothelial cells (Figs 6A and S6A). Also, kininogen and vitronectin sightly enhanced the endocytosis of wild-type *C. albicans*, which formed hyphae on the endothelial cells, and largely rescued the endocytosis and adherence defects of the invasin-deficient *als3Δ/Δ ssa1Δ/Δ* mutant (Figs 6B and S6B).

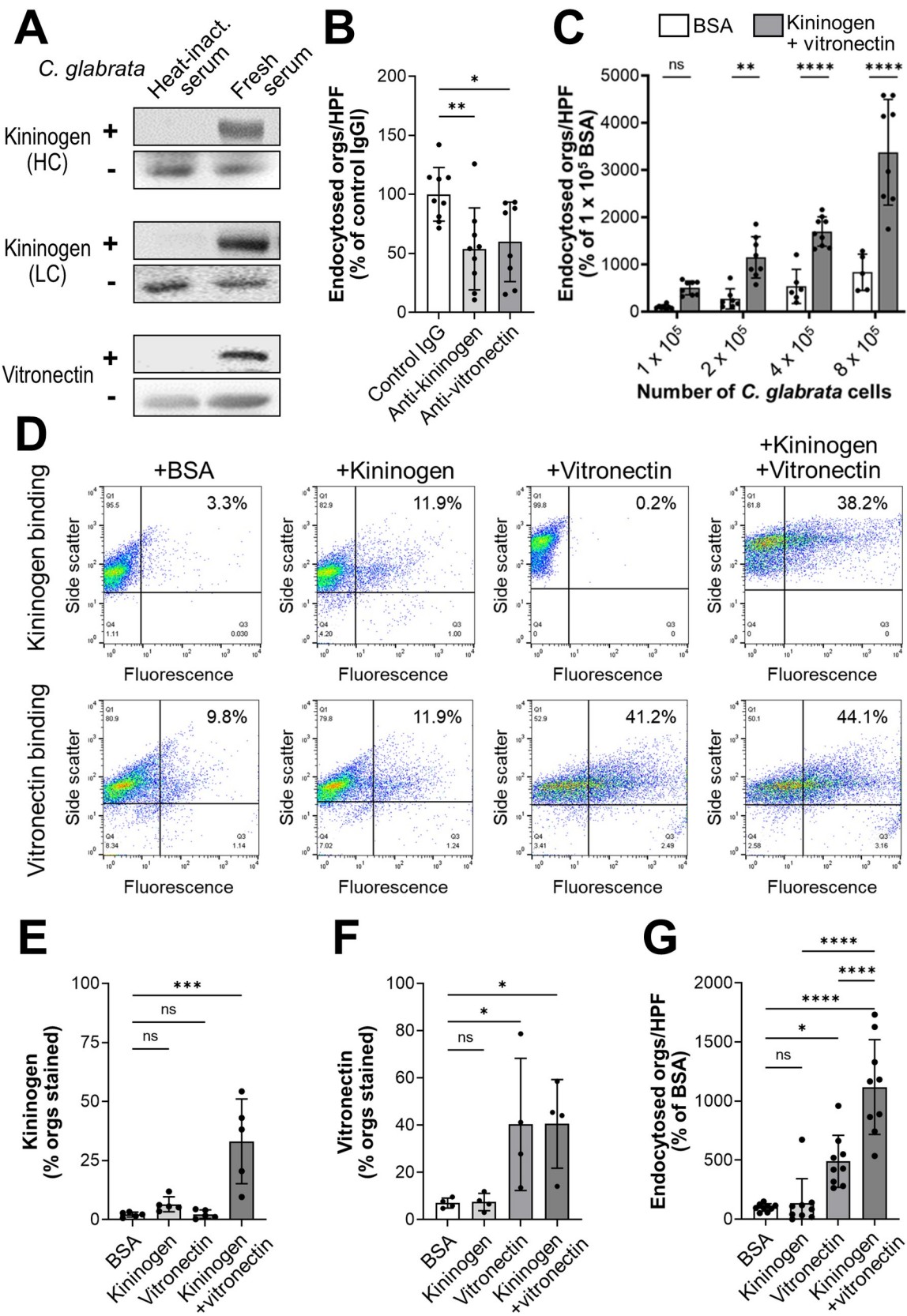

**Fig 5. High molecular weight kininogen and vitronectin function as bridging molecules.** (A) Western blots showing that the heavy chain (HC) and light chain (LC) of high molecular weight kininogen and vitronectin bind to *C. glabrata* cells that have been incubated in fresh human serum. In each pair of blots, the upper panel shows the proteins that were eluted from *C. glabrata* and lower panel shows the proteins present in serum in the absence of *C. glabrata*. (B) Effects of anti-kininogen and anti-vitronectin antibodies on the endocytosis of serum-coated *C. glabrata* by human endothelial cells. (C) Endocytosis of the indicated number of *C. glabrata* cells that had been coated with either BSA or kininogen and vitronectin. (D) Flow cytometric detection of the binding of kininogen (top row) and vitronectin (bottom row) to *C. glabrata* cells that had been incubated for 1 h with BSA without kininogen or vitronectin, kininogen alone, vitronectin alone, or kininogen and vitronectin. Numbers in the upper right hand corner indicate the percentage of positive cells. Results are representative of 5 (kininogen) or 4 (vitronectin) separate experiments, each of which analyzed 10,000 cells. (E-F) Summary of combined flow cytometry results showing the binding of kininogen (E) and vitronectin (F) to *C. glabrata* cells. (G) Endocytosis of *C. glabrata* cells that had been coated with the indicated proteins. Data in (B), (C), and (G) are the mean ± SD of 3 experiments each performed in triplicate. Orgs/HPF, organisms per high power field; ns, not significant; $^{*}P < 0.05$, $^{**}P < 0.01$, $^{***}P < 0.001$, $^{****}P < 0.0001$ by ANOVA with the Dunnett's test for multiple comparisons.

When wild-type *C. albicans* is endocytosed by endothelial cells, it damages these cells, likely by releasing candidalysin into the invasion pocket [6,29–31]. We tested whether coating the *als3Δ/Δ ssa1Δ/Δ* mutant with kininogen and vitronectin would restore is capacity to damage endothelial cells. While organisms coated with BSA caused minimal endothelial cell damage, cells coated with kininogen and vitronectin induced significantly greater damage (Fig 6C). These results indicate that in the absence of invasins, bridging molecules can enhance the endocytosis of *C. albicans* hyphae, leading to subsequent endothelial cell damage.

As we had observed that fresh serum increased the endocytosis of species of *Candida*, other than *C. albicans*, we investigated whether human kininogen and vitronectin functioned as bridging molecules for these organisms. We found that these proteins significantly increased the endocytosis of *C. parapsilosis*, *C. tropicalis*, and *C. krusei*, but not *C. auris*. (Fig 6D). Kininogen and vitronectin also increased the endothelial cell adherence of *C. parapsilosis* and *C. krusei* (S6C Fig). Next, we tested whether the enhanced endocytosis of these organisms by kininogen and vitronectin would result in endothelial cell damage. To increase the sensitivity of the experiment, we increased the inoculum and extended the incubation period to 6 hr. None of these organisms caused detectable damage to the endothelial cells (S6D Fig), indicating that induction of endocytosis alone is not sufficient for these species of *Candida* to cause significant endothelial cell damage.

To investigate the specificity of kininogen and vitronectin in inducing endocytosis, we compared *C. glabrata* with *S. cerevisiae*, which is not endocytosed when coated with human serum (S1G Fig). We found that there was minimal endocytosis or adherence of *S. cerevisiae* cells coated with kininogen and vitronectin (Figs 6E and S6E), indicating that kininogen and vitronectin do not act as bridging molecules for this organism.

To investigate which endothelial cell receptor was responsible for interacting with each bridging molecule, we tested the inhibitory effects of specific antibodies directed against gC1qR and αv integrins. When *C. glabrata* cells were incubated with vitronectin alone, endocytosis was significantly inhibited by antibodies against integrins αvβ3 and αvβ5, but not by the anti-gC1qR antibody (Fig 6F). None of these antibodies significantly reduced the adherence of vitronectin-coated organisms (S6F Fig). When the organisms were incubated with vitronectin and kininogen, endocytosis was inhibited by both the anti-gC1qR antibody and the anti-αv integrin antibodies (Fig 6G). The combination of all 3 antibodies inhibited endocytosis in an additive manner and also inhibited adherence (S6G Fig). Taken together, these data support the model that vitronectin likely binds first to the fungal surface where it is recognized mainly by integrins αvβ3 and αvβ5 (Fig 7). Binding of vitronectin enables kininogen to bind to fungal cell surface, and the vitronectin-kininogen complex is recognized by both gC1qR and the αv integrins, leading to the strong adherence and subsequent endocytosis of the organism.

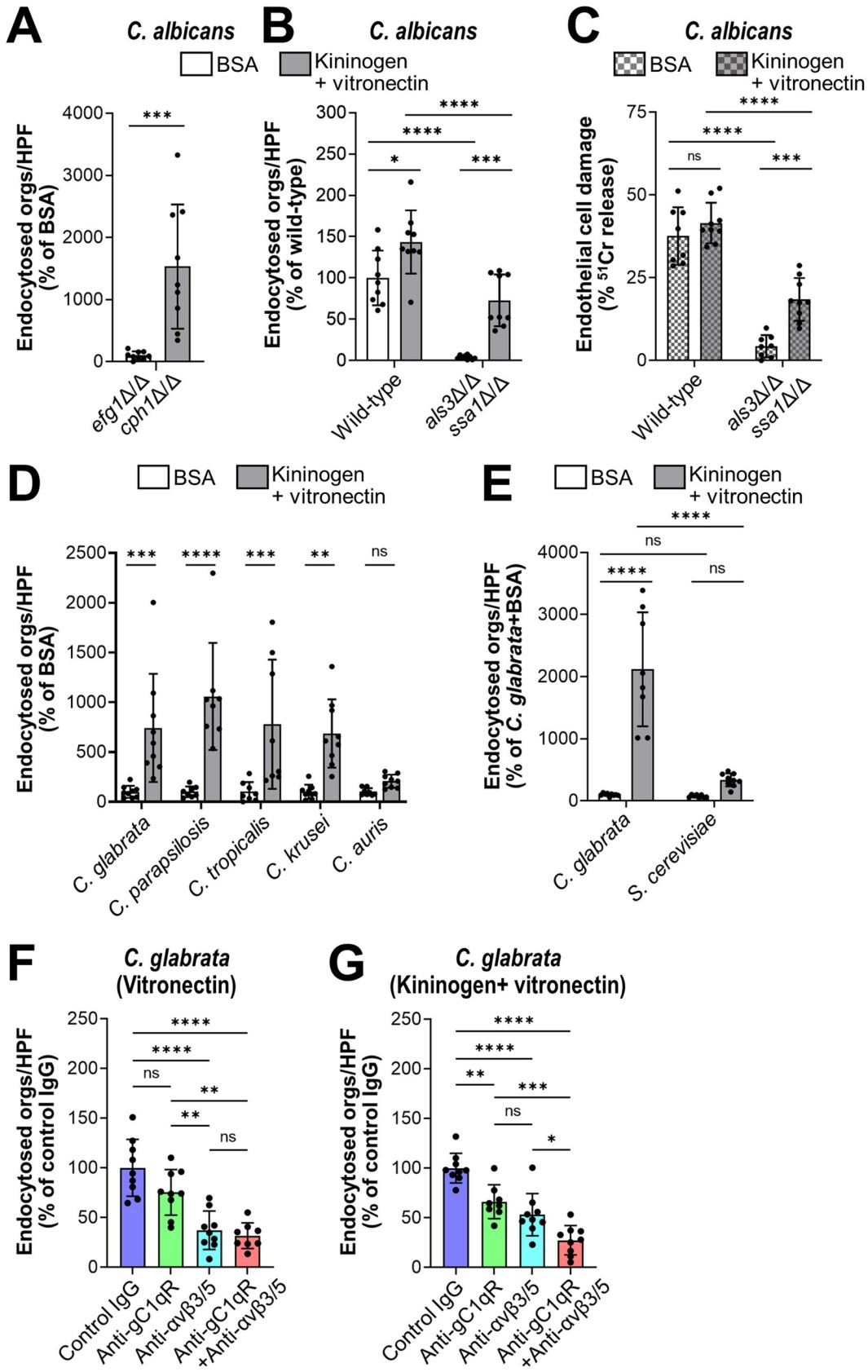

**Fig 6. Kininogen and vitronectin interact with gC1qR and αv integrins to induce endocytosis by human endothelial cells.** (A and B) Effects of BSA or kininogen and vitronectin on the endocytosis of live cells of the indicated strains of *C. albicans* after 90 min. (C) Effects of BSA or kininogen on endothelial cell damage caused by the indicated strains of *C. albicans*. (D) Kininogen and vitronectin increase endothelial cell endocytosis of the indicated *Candida spp*. (E) Kininogen and vitronectin do not enhance the endocytosis of *S. cerevisiae*. (F and G) Inhibition of endocytosis of *C. glabrata* coated with either vitronectin alone (E) or vitronectin and kininogen (F) by antibodies against gC1qR (clone 74.5.2) and/or integrins αvβ3 and αvβ5. Data are the mean ± SD of 3 experiments each performed in triplicate. Orgs/HPF, organisms per high power field; ns, not significant; $^*P < 0.05$, $^{**}P < 0.01$, $^{***}P < 0.001$, $^{****}P < 0.0001$ by ANOVA with the Dunnett's test for multiple comparisons.

## Expression of human gC1qR and αv integrins on mouse endothelial cells enhances bridging molecule mediated endocytosis

Next, we investigated whether mouse serum bridging molecules also mediated the endocytosis of *C. glabrata* by comparing the capacity of mouse and human serum to mediate endocytosis by human endothelial cells. To maximize endocytosis, we incubated the organisms in 100% serum. We observed that after 45 min, mouse serum enhanced the endocytosis of *C. glabrata* by human endothelial cells, but to a lesser extent than human serum (Fig 8A). Mouse serum also increased adherence to human endothelial cells, but not as much as human serum (S7A Fig). These differences in endocytosis and adherence persisted even when the incubation period was increased to 3 h (Figs 8B and S7B). These results indicate that while mouse serum proteins can function as bridging molecules between *C. glabrata* and human endothelial cells, they are less effective than human serum proteins.

To investigate whether serum bridging molecules could mediate the endocytosis of *C. glabrata* by mouse endothelial cells, we obtained primary mouse kidney and liver endothelial cells and tested their capacity to endocytose *C. glabrata* cells that had been coated with either human or mouse serum. We found that there was minimal endocytosis and adherence of

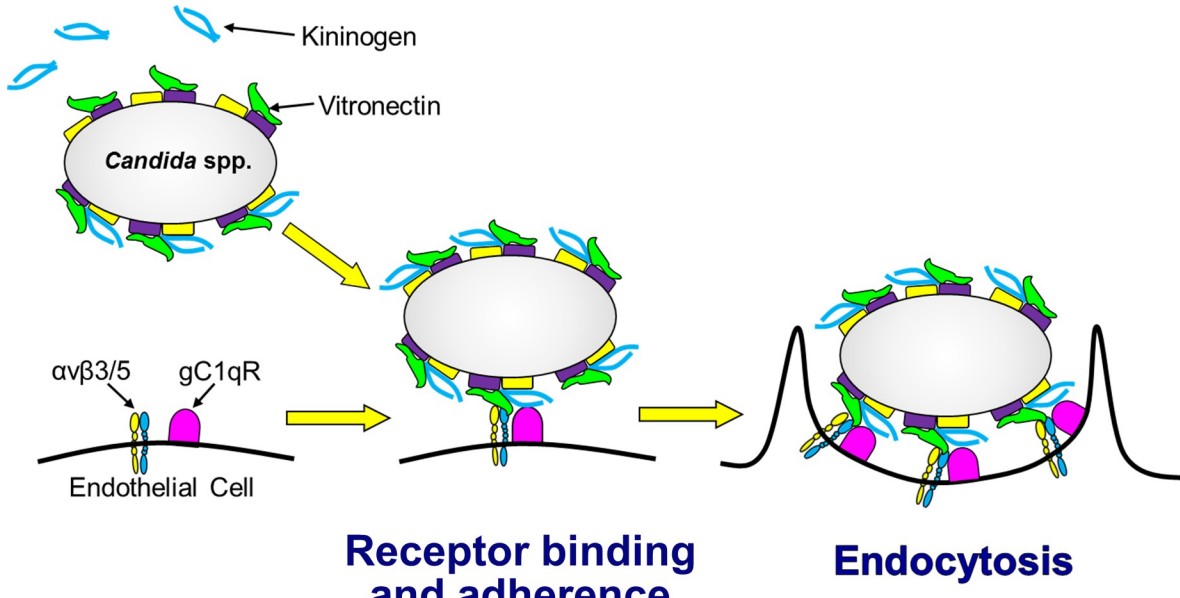

**Fig 7. Model of how kininogen and vitronectin function as bridging molecules that bind to *Candida spp*. and induce endocytosis by human endothelial cells.** Vitronectin binds to the surface of the organism, which enhances the binding of kininogen. When the organism comes in contact with the endothelial cell, vitronectin interacts mainly with the integrins αvβ3 and αvβ5 and whereas the vitronectin-kininogen complex interacts with both the αv integrins and gC1qR on the endothelial cells surface. These interactions mediate the adherence of the fungus to the endothelial cells and induce the endothelial cells to endocytose the organism.

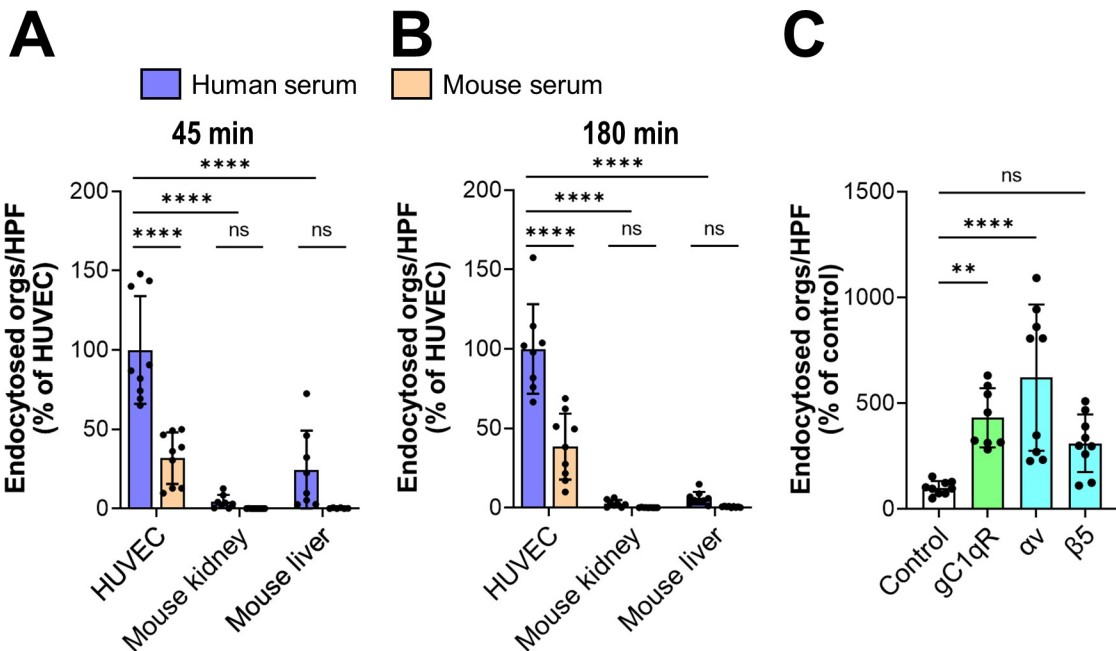

**Fig 8. Mouse endothelial cells poorly support bridging molecule-mediated endocytosis.** (A and B) Endocytosis of *C. glabrata* coated with either human or mouse serum by the indicated endothelial cells after 45 min (A) and 180 min (B). (C) Endocytosis of *C. glabrata* coated with fresh human serum by mouse liver endothelial cells expressing human gC1qR, integrin αv, or integrin β5. Data are the mean ± SD of 3 experiments each performed in triplicate. HUVEC, human umbilical vein endothelial cell; orgs/HPF, organisms per high power field; ns, not significant; $^{**}P < 0.01$, $^{****}P < 0.0001$. $^{***}P < 0.001$, $^{****}P < 0.0001$ by ANOVA with the Dunnett's test for multiple comparisons.

organisms coated with human or mouse serum by mouse endothelial cells after both 45 min and 180 min (Figs 8A and 8B and S7A and S7B). To verify that human endothelial cells other than those obtained from umbilical cord veins were able to endocytose serum coated organisms, we tested a Tert-immortalized human microvascular endothelial (TIME) cell line. *C. glabrata* cells coated with either human serum or human kininogen and vitronectin were endocytosed by and adhered to the TIME cells more than human umbilical vein endothelial cells (S7C–S7F Fig). Collectively, these data indicate that while both mouse and human serum proteins can function as bridging molecules between *C. glabrata* and human endothelial cells, mouse endothelial cells have very limited capacity to endocytose organisms coated with serum from either mice or humans.

We considered the possibility that the inability of mouse endothelial cells to endocytose serum coated *C. glabrata* was due to difference between the receptors on mouse vs. human endothelial cells. To evaluate the possibility, we used lentivirus to transduce primary mouse liver endothelial cells with human *C1QBP* (gC1qR), *ITGAV* (integrin αv), or *ITGB5* (integrin β5). Control cells were transduced with lentivirus encoding GFP. The expression of the human proteins by the transduced endothelial cells was verified by Western blotting (S7G Fig). Endothelial cells that expressed human gC1qR and integrin αv endocytosed significantly more serum-coated organisms than did the control endothelial cells (Fig 8C). Endothelial cells that expressed human gC1qR, integrin αv, and integrin β5 also had enhanced *C. glabrata* adherence (S7H Fig). These data demonstrate that human gC1qR and integrin αv mediate the endocytosis and adherence of serum-coated *C. glabrata*. They also suggest that these human receptors are functionally different from their mouse counterparts.

## Discussion

In this study, we sought to elucidate how species of *Candida* that do not form true hyphae are able to invade vascular endothelial cells. Using *C. glabrata* as a representative fungus that grows only as yeast within the human host, we determined that proteins present in human serum act as bridging molecules between the fungus and human endothelial cells and induce the adherence and subsequent endocytosis of the organism. The data presented here indicate that binding of vitronectin to the fungal surface facilitates the subsequent binding of kininogen. Vitronectin interacts mainly with the integrins αvβ3 and αvβ5 endothelial cells, and the kininogen-vitronectin complex also interacts with gC1qR. The binding of these serum proteins to their respective receptors causes the organism to adhere to endothelial cells and stimulates its subsequent endocytosis (Fig 7). Not only did kininogen and vitronectin act as bridging molecules for *C. glabrata*, but they also mediated endothelial cell endocytosis of yeast-locked and invasin-deficient *C. albicans* mutants and other medically important species of *Candida*, including *C. tropicalis*, *C. parapsilosis*, and *C. krusei*. Although serum enhanced the endocytosis of *C auris*, kininogen and vitronectin did not, suggesting that other serum proteins must function as bridging molecules for this organism. Also, serum bridging molecules did not induce the endocytosis of *S. cerevisiae*, indicating that bridging molecule-mediated endocytosis is not a general property of yeast. Taken together, these results indicate that invasion of vascular endothelial cells via bridging molecule-mediated endocytosis is a pathogenic strategy shared by many medically important *Candida* spp.

Vitronectin, which is bound by integrins αvβ3 and αvβ5, is known to function as a bridging molecule that mediates adherence to respiratory epithelial cells of a variety of bacteria, including nontypeable *Haemophilus influenzae*, *Moraxella catarrhalis*, group A streptococci, and *Pseudomonas aeruginosa* (reviewed in [32]). In addition to mediating adherence, vitronectin induces the internalization of *Neisseria gonorrhoeae* and *Pseudomonas fluorescens* by epithelial cells [33,34]. *C. albicans*, C. parapsilosis and C. tropicalis have been shown to bind to fluid phase vitronectin [35], and the binding of *C. albicans* to vitronectin mediates adherence to keratinocytes [36]. Our findings demonstrate that *C. glabrata* and *C. krusei* are additional *Candida* spp. that bind to vitronectin. More importantly, we show that vitronectin acts as a bridging molecule that, in conjunction with kininogen, mediates the endocytosis of these organisms by human endothelial cells.

Studies of the interaction of kininogen with microbial pathogens have focused mainly on its proteolytic cleavage to release bradykinin and other fragments with antimicrobial activity. Kininogen has been found to bind to *S. aureus*, *Salmonella typhimurium*, and *Bacteroides* spp. [37,38]. Rapala-Kozik et al., have determined that virtually all medically important *Candida spp.* bind kininogen [39–41]. In contrast to the results shown here, they found that kininogen could bind to the fungus in the absence of additional serum proteins, whereas we found that there was minimal binding of kininogen to *C. glabrata* unless vitronectin was present. The likely explanation for these divergent results is that the other investigators used a more sensitive assay that was able to detect the binding of even low amounts of kininogen to the fungal surface. Nevertheless, our results indicate that vitronectin dramatically increases the amount of kininogen that binds to *C. glabrata* and enables kininogen to function as a bridging molecule that enhances fungal endocytosis.

Although the function of kininogen as a bridging molecule between microbial pathogens and host cells has not been appreciated previously, it is known that kininogen can bind to glycoprotein 1b on platelets and integrin αMβ2 on neutrophils to enhance the co-adherence of these two cells [42]. We determined that unlike platelets and neutrophils, endothelial cells bind kininogen via gC1qR, a result that has been reported by others [43]. gC1qR has also been

found to be a receptor for *Listeria monocytogenes* that mediates the internalization of this organism. However, the bacterium binds directly to gC1qR, and this interaction can be blocked by both C1q and monoclonal antibody 60.11, which is directed against the C1q binding site of gC1qR [44]. By contrast, we found that while monoclonal antibody 60.11 did not inhibit bridging molecule-mediated endocytosis of *C. glabrata*, monoclonal antibody 74.5.2, which is directed against the kininogen binding site of gC1qR, was highly inhibitory. These results support the model that when kininogen is bound to the surface of *Candida* spp., it interacts with gC1qR on endothelial cells and stimulates the endocytosis of the organisms.

Although gC1qR was required for the maximal endocytosis of serum-coated *C. glabrata*, this protein lacks a transmembrane sequence and thus likely induces endocytosis by associating with other endothelial cell surface proteins. For example, we recently found that in oral epithelial cells, gC1qR forms a complex with the epidermal growth factor receptor (EGFR), and this association is necessary for *C. albicans* to activate EGFR and induce its own endocytosis [45]. Because endothelial cells express little to no EGFR, it is likely that gC1qR associates with other endothelial cell surface proteins, possibly integrins αvβ3/5 [46].

Patients with hematogenously disseminated candidiasis due to *C. glabrata*, *C. tropicalis*, *C. krusei*, and *C auris* have at least as high mortality as those who are infected with *C. albicans* [2,16,47]. These data suggest that in humans, these different species of *Candida* have similar virulence. In immunocompetent mice, *C. albicans* is highly virulent, and most wild-type strains are capable of causing a lethal infection. By contrast, intravenous infection of immunocompetent mice with *C. glabrata*, *C. tropicalis*, *C. krusei*, and *C. auris* induces minimal mortality even when high inocula are used [48,49]. Thus, in mice, these species of *Candida* have greatly attenuated virulence. A possible explanation for this discrepancy is that *C. albicans* is able to form hyphae that express invasins such as Als3 and Ssa1 that interact directly with endothelial cell receptors and induce endothelial cell endocytosis. By contrast, other species of *Candida*, such as *C. glabrata* invade endothelial cells by bridging molecule-mediated endocytosis, a process that occurs inefficiently in mouse endothelial cells in culture.

The results of the in vitro studies support this concept. *C. albicans* was endocytosed avidly by human endothelial cells in the absence of serum and coating the organism with serum only increased endocytosis slightly. By contrast, *C. glabrata* was poorly endocytosed in the absence of serum, and coating the organisms with serum dramatically increased endocytosis. Both mouse and human proteins increased endocytosis by human cells, indicating that mouse serum proteins can function as bridging molecules, albeit not as well as human proteins. Importantly, *C. glabrata* was poorly endocytosed by mouse liver and kidney endothelial cells when it was coated with either mouse or human serum. When the mouse endothelial cells were engineered to express human gC1qR or integrin αv, they gained the capacity to endocytose serum-coated *C. glabrata*. These data indicate that a key difference between mouse and human endothelial cells in vitro is that mouse gC1qR and integrin αv do not support bridging molecule-mediated endocytosis of *C. glabrata*. The role of bridging molecules in mediating endothelial cell invasion in vivo is likely more complex because of the multitude of cell types and receptors that interact with the fungus during mammalian infection.

Although the mouse model of disseminated candidiasis is an excellent model for investigating antifungal efficacy and many aspects of fungal pathogenicity, our results suggest that this model is not optimal for investigating how *C. glabrata* and possibly other *Candida* spp. other than *C. albicans* disseminate hematogenously because mouse endothelial cells do not support bridging molecule-mediated vascular invasion. Even though mice inoculated intravenously with these organisms still contain some fungal cells in their tissues, we speculate that the organisms must egress from the vasculature by another mechanism(s) that has less pathogenic impact. This possibility is currently being investigated.

The results presented here indicate that many medically important species of *Candida* can utilize serum proteins as bridging molecules to induce their own endocytosis by human vascular endothelial cells. Because this mechanism is shared by multiple *Candida spp.*, it represents a promising therapeutic target for preventing or ameliorating hematogenously disseminated candidiasis.

## Methods

### Ethics statement

All animal work was approved by the Institutional Animal Care and Use Committee (IACUC) of the Lundquist Institute for Biomedical Innovation at Harbor-UCLA Medical Center. The human subjects research was approved by the IRB of the Lundquist Institute for Biomedical Innovation at Harbor-UCLA Medical Center under protocol 30636-01R. Written informed consent was obtained prior to phlebotomy.

### Serum and plasma

After obtaining informed consent, blood was collected by venipuncture from healthy volunteers. Blood was also collected from anesthetized Balb/C mice by cardiac puncture. To obtain serum, the blood was allowed to clot at room temperature for 30 min and then centrifuged at 2000 rpm for 10 min at 4˚C. After collecting the serum, samples from individual donors were pooled and stored in aliquots in liquid nitrogen. To make heat-inactivated serum, the fresh serum was incubated at 56˚C for 1 h and stored in aliquots in liquid nitrogen.

To obtain plasma, fresh human blood was transferred to 4 ml vacutainer tubes containing 7.2 mg of $K_2$EDTA (BD, Inc.). The tubes were then centrifuged at 2000 rpm for 10 minutes at 4˚C, after which the plasma was collected, pooled, aliquoted, and stored in liquid nitrogen. Heat-inactivated plasma was made by incubating fresh plasma at 56˚C for 1 hr.

### Host cells, fungal strains and growth conditions

Human umbilical vein endothelial cells were isolated from umbilical cords and grown as described and used at passage one [30,50]. Mouse kidney endothelial cells (Cell Biologics), mouse liver endothelial cells (Cell Biologics), and hTert-immortalized human microvascular endothelial cells (American Type Culture Collection) were purchased and grown according to the suppliers' instructions.

The fungal strains used in this work are listed in S1 Table. For use in the experiments, the organisms were grown overnight in yeast extract peptone dextrose (YPD) broth at 30˚C in a shaking incubator. They were harvested by centrifugation, washed twice with PBS and enumerated with a hemacytometer as previously described [30]. To produce killed organisms, cells of *C. albicans* strain DIC185 were pelleted by centrifugation and then resuspended in 100% methanol for 2 min. The killed organisms were recovered by centrifugation and washed two times with PBS. Fungal killing was verified by plating a sample of the cells onto YPD agar.

Strain DSC10 was constructed by plating strain CAN34 (*efg1*Δ/Δ *cph1*Δ/Δ) on minimal medium containing 5-fluororotic acid. The resulting Ura- strain was transformed with a PstI/NotI-digested fragment of pBSK-URA3 [51] to restore the *URA3-IRO1* locus. Proper integration was verified by PCR.

### Coating fungal cells with bridging molecules

To coat the organisms with serum or plasma, approximately $5 \times 10^7$ fungal cells were mixed with either RPMI 1640 medium alone (Irvine Scientific) or RPMI 1640 medium containing

20% fresh or heat-inactivated human serum and then incubated for 1 h at 37˚C in a shaking incubator. In some experiments, the human serum was replaced with human plasma to which $CaCl_2$ was added to reverse the effects of the EDTA. In experiments comparing mouse with human serum, the organisms were incubated with either 100% mouse or human serum. After coating, the fungal cells were washed twice with PBS, diluted, and counted for use in the assays described below.

To coat the organisms with bridging molecules, approximately $2x10^7$ fungal cells were incubated with human kininogen (10 μg/ml; Molecular Innovations, Inc., Cat. # HK-TC) and/or human vitronectin (30 μg/ml; Molecular Innovations Inc., Cat. # HVN-U) in RPMI 1640 medium supplemented with 50 μM $ZnCl_2$ and 3 μM $CaCl_2$. Control cells were incubated with BSA (Sigma-Aldrich). The cells were incubated for 1 h at 37˚C in a shaking incubator and processed as described above.

## Confocal microscopy

The confocal microscopy was performed as previously described [6]. Briefly, endothelial cells were grown to confluency on fibronectin-coated glass coverslips and then infected with $3x10^5$ *C. glabrata* cells. After 45 min, the cells were fixed with 3% paraformaldehyde and blocked with 5% goat serum containing 0.05% Triton X-100. The cells were incubated with Alexa Fluor 568-labeled phalloidin (Thermo Fisher Scientific, Cat. #A12380), rabbit anti-gC1qR antibody (Santa Cruz Biotechnology, Cat. #sc-48795), anti-integrin αvβ3 monoclonal antibody (Millipore-Sigma, clone LM609, Cat. # MAB1976), or anti-integrin αvβ5 monoclonal antibody (Millipore-Sigma, clone P1f6, Cat. # MAB1961). The *C. glabrata* cells were labeled with calcofluor white. After extensive rinsing, the cells were incubated with the appropriate Alexa Fluor labeled secondary antibody (Thermo Fisher Scientific, Cat. #A-11031 or A-11034), rinsed, and then imaged by confocal microscopy. Consecutive z-stacks were combined to create the final images.

## Endocytosis assay

The endocytosis of the various organisms by endothelial cells was determined by our standard differential fluorescence assay as described previously [6,52]. Briefly, endothelial cells grown on fibronectin-coated glass coverslips were incubated with $10^5$ fungal cells in 5% $CO_2$ at 37˚C for 45 or 180 min. Next, the cells were fixed in 3% paraformaldehyde, and the non-endocytosed organisms were stained with an anti-*Candida* antibody (Meridian Life Science, Cat. # B65411R) that had been conjugated with Alexa Fluor 568 (Thermo Fisher Scientific, Cat. # A-10235). After rinsing the cells extensively with PBS, the endothelial cells were permeabilized in 0.05% Triton X-100 (Sigma-Aldrich), and the cell-associated organisms were stained with the anti-*Candida* antibody conjugated with Alexa Fluor 488 (Thermo Fisher Scientific). The coverslips were mounted inverted on microscope slides and viewed with an epifluorescent microscope. The number of endocytosed organisms was determined by scoring at least 100 organisms per slide. Each experiment was performed at least three times in triplicate.

The effects of depolymerizing microfilaments on endocytosis was determined by incubating the endothelial cells with 0.4 μM cytochalasin D (Sigma-Aldrich) for 45 min prior to infection. Control endothelial cells were incubated in the diluent (0.1% DMSO) in parallel. The cytochalasin D and DMSO remained in the medium for the duration of the infection. To determine the effects of blocking endothelial cell receptors on endocytosis, the endothelial cells were incubated with anti-gC1qR antibodies (Santa Cruz Biotechnology, clone 74.5.2 Cat. # sc-23885 and Abcam, clone 60.11, Cat. # ab24733), anti-αvβ3 antibody (Millipore Sigma, clone LM609, Cat. # MAB1976), αvβ5 (Millipore Sigma, clone P1F6, Cat. # MAB1961), or a combination of

antibodies, each at 10 μg/ml. Control cells were incubated in the same concentration of mouse IgG (R&D Systems, clone 11711; # MAB002). The endothelial cells were incubated with the antibodies for 1 h prior to infection and the antibodies remained in the medium for the duration of infection.

To determine the effects of inhibiting bridging molecules on endocytosis, *C. glabrata* cells were coated with 20% heat-inactivated or fresh serum in the presence of an anti-kininogen antibody (Santa Cruz Biotechnology, clone 2B5, Cat. # sc-23914), an anti-vitronectin antibody (Millipore Sigma, clone 8E6(LJ8), Cat. # MAB88917), or an isotype control IgG, each at 10 μg/ml. The organisms were then washed twice with PBS, counted, and used in the endocytosis assay.

Because of day to day variation in the number of organisms that were endocytosed by and cell-associated with endothelial cells from different umbilical cords, the data were normalized to control cells. The mean and standard deviation of the raw data from each experiment are provided in S2 Table.

## Protein purification and Western blotting

Endothelial cell membrane proteins were isolated using glucopyranoside according to our previously described method[6-8]. To pull down endothelial cell proteins that bound to serum-coated *C. glabrata*, 8x10$^8$ organisms that had been coated with fresh or heat-inactivated serum were incubated with 1 mg of endothelial cell membrane proteins on ice for 1 hr. Unbound proteins were removed by rinsing with 1.5% glucopyranoside, after which the bound proteins were eluted with 6M Urea. Samples were added to SDS-PAGE sample buffer, heated to 90˚C for 5 min, and then separated by SDS-PAGE. To detect gC1qR that was associated with *C. glabrata* cells that had been coated with fresh or heat-inactivated serum, Western blotting was performed with the anti-gC1qR antibody (clone 74.5.2).

To detect serum proteins that bind to *C. glabrata*, 1x10$^8$ *C. glabrata* cells were incubated with 20% fresh or heat-inactivated serum in RPMI 1640 medium for 1 hr at 37˚C. Unbound serum proteins were removed by rinsing the cells twice with PBS, after which bound serum proteins were eluted with 2M HCl, pH 2.0 and immediately neutralized with Tris buffer, pH 8.0. The proteins were separated by SDS-PAGE and Western blotting using an anti-kininogen heavy chain antibody (Santa Cruz Biotechnology, clone 2B5, Cat. # sc-23914), anti-light chain antibody (Santa Cruz Biotechnology, clone 14J09, Cat. # sc-80524), and anti-vitronectin antibody (clone 8E6(LJ8)) was performed to detect kininogen and vitronectin that had been eluted from *C. glabrata*.

## siRNA

Knockdown of endothelial cell surface proteins was accomplished using siRNA. The endothelial cells were transfected with gC1qR siRNA (Santa Cruz Biotechnology, Cat. # sc-42880), integrin α5 siRNA (Santa Cruz Biotechnology, Cat. # sc-29372), integrin αv siRNA (Santa Cruz Biotechnology, Cat. # sc-29373), or scrambled control siRNA (Qiagen, Cat. # 1027281) using Lipofectamine 2000 (Thermo Fisher Scientific) following the manufacturer's instructions. After 48 hr, the transfected endothelial cells were infected with serum coated *C. glabrata* and the number of endocytosed organisms was determined. Knockdown of each protein was verified by immunoblotting with antibodies against gC1qR (clone 74.5.2), integrin α5 (Millipore-Sigma, Cat. # AB1928), integrin αv (Santa Cruz Biotechnology, clone H-2, Cat # sc-376156), integrin β3 (Santa Cruz Biotechnology, clone B-7, Cat. # sc-46655), integrin β5 (Santa Cruz Biotechnology, clone F-5, Cat. # sc-398214), or actin (Millipore-Sigma, clone C4, Cat. # A5441-100UL)

## Flow cytometry

The binding of kininogen and vitronectin to *C. glabrata* and methanol-fixed *C. albicans* was analyzed by a modification of a previously described method [53]. Fungal cells were incubated with kininogen that had been labeled with Alexa Fluor 568 (Thermo Fisher Scientific, Cat. #A20184) and/or unlabeled vitronectin, both at a final concentration of 30 μg/ml, for 1 h at 37˚C. Control cells were incubated in a similar concentration of BSA. The unbound proteins were removed by washing the cells twice with PBS. Next, the cells were incubated with the anti-vitronectin antibody followed by the Alexa Fluor 488-labeled goat anti-mouse secondary antibody. The fluorescence of the cells was then quantified using a Becton Dickinson FACScalibur flow cytometer, analyzing 10,000 cells per sample using the FlowJo software.

The potential binding the anti-αvβ3 and anti-αvβ5 antibodies to *C. glabrata* was determined by incubating *C. glabrata* cells with each antibody at a final concentration of 10 μg/ml, followed by the Alexa Fluor 488-labeled goat anti-mouse secondary antibody The fluorescence of the cells was then quantified by flow cytometer, analyzing 10,000 cells per sample.

## Endothelial cell damage assay

The capacity of wild-type (CAI4-URA) and *als3Δ/Δ ssa1Δ/Δ C. albicans* strains to damage human umbilical vein endothelial cells was determined using our previously described $^{51}$Cr release assay [6]. Endothelial cells were grown in a 96-well tissue culture plate containing detachable wells and loaded with $^{51}$Cr. The *C. albicans* were coated with either BSA or kininogen and vitronectin and rinsed, after which 4 x 10$^4$ fungal cells were added to individual wells of endothelial cells. After incubation for 3 h, the medium above the cells was aspirated and the wells were detached from each other. The amount of $^{51}$Cr released into the medium and remaining in the endothelial cells was determined using a gamma counter.

When the *C. glabrata*, *C. tropicalis*, *C. parapsilosis*, *C. krusei*, and *C. auris* strains were tested in the damage assay, they were processed similarly to the *C. albicans* cells except that the inoculum was increased to 2 x 10$^5$ cells per well and the incubation period was increased to 6 h.

## Lentivirus packaging and host cell transduction

The transfer vectors (pLenti-EF1A-EGFP-Blast, pLenti-EF1A-hITGAV-NEO, pLenti-EF1A-hITGB5-NEO or pLenti-EF1A-hC1QBP-Blast) were constructed by cloning eGFP, hC1QBP [NM_001212.4], hITGAV [NM_002210], or hITGB5 [NM_002213] into pLenti-Cas9-Blast (Addgene; # 52962) or pLenti-Cas9-NEO at the BamHI and XbaI sites. The virus was produced by transfecting HEK293T cells with plasmid psPAX2 (Addgene; # 12260), plasmid pCMV-VSVG (Addgene; # 8454), and transfer vector at 1:1:1 molar ratio using the X-treme-GENE 9 DNA transfection reagent (Sigma-Aldrich; # 6365787001) according to the manufacturer's instructions. The supernatant containing the virus was collected at 60 h post-transfection, passed through a 0.45um PVDF filter and stored at 4˚C (short-term) or -80˚C (long-term).

For transduction, mouse primary liver endothelial cells were seeded into a 6-well plate in mouse endothelial cell medium (Cell Biologics, Inc. # M1168). The cells were transduced with lentivirus in the presence of 8 μg/ml polybrene (Santa Cruz Biotechnology; #SC134220), centrifuged at 1000*g* for 30 min, and then incubated at 37˚C in 5% CO$_2$ overnight. The next morning, the cells were transferred to 10 cm diameter tissue culture dishes. For the cells transduced with hC1QBP, 10 μg/ml of blasticidin (Gibco; # A1113903) was added to the medium 2 d post transduction to select for transduced cells and selection was maintained for 7 d. Expression of eGFP was determined by fluorescent microscopy and expression of gC1qR, integrin αv, and integrin β5 were verified via immunoblotting of whole cell lysates with an anti-gC1qR

antibody (clone 60.11), anti- integrin αv antibody (MilliporeSigma; #AB1930), and anti- integrin β5 antibody (My Biosource, Inc; # MBS617750). Total loading was determined by immunoblotting with an anti-GAPDH antibody (Cell Signaling; # 5174).

## Statistical analysis

All data were analyzed using Prism (GraphPad). Differences among experimental groups were analyzed by one-way analysis of variance followed by pair-wise analysis with Dunnett's multiple comparison test. When a single pair of data was analyzed, the 2-way student's t-test assuming unequal variance was used. $P$ values $\leq 0.05$ were considered to be significant.

## Supporting information

**S1 Fig. Bridging molecules mediate the endocytosis of live *C. albicans* and *C. glabrata* yeast, but not *S. cerevisiae* by human endothelial cells.** (A and B) Endocytosis (A) and cell-association (B) of live cells of a *C. albicans egf1Δ/Δ cph1Δ/Δ* mutant by human umbilical vein endothelial cells. (C and D) Effects of fresh and heat-inactivated serum on the endocytosis (C) and cell-association of the indicated strains of live *C. glabrata*. (E and F) Effects of fresh human serum and plasma on the endocytosis (E) and cell-association (F) of live *C. glabrata*. (G and H) Endocytosis (G) and cell-association (H) of live *C. glabrata* and *S. cerevisiae*. Results are the mean ± SD of 3 independent experiments, each performed in triplicate. Orgs/HPF, organisms per high-power field; ns, not significant; $^{**}P < 0.01$, $^{***}P < 0.001$, $^{****}P < 0.0001$ by Student's t-test (A and B) or ANOVA with the Dunnett's test for multiple comparisons (C-H).
(PDF)

**S2 Fig. Western blot showing effects of gC1qR siRNA on the levels of the indicated human endothelial cell proteins.**
(PDF)

**S3 Fig.** (A) Western blot showing effects of integrin αv siRNA on the levels of the indicated endothelial cell proteins. (B) Antibodies against gC1qR, integrin αvβ3, and integrin αvβ5 do not bind to *C. glabrata*. Flow cytometric analysis of *C. glabrata* cells that were incubated with antibodies against gC1qR (clone 74.5.2), integrin αvβ3, and integrin αvβ5. They were also incubated with control IgG and with a polyclonal anti-*Candida* antibody. Each histogram shows the analysis of $10^4$ cells.
(PDF)

**S4 Fig.** (A and B) Combined effects of antibodies against gC1qR, integrin αvβ3, and integrin αvβ5 on the endocytosis (A) and cell association (B) of *C. glabrata*. Data are the mean ± SD of 3 experiments each performed in triplicate. HI, heat-inactivated; Orgs/HPF, organisms per high power field; ns, not significant; $^{**}P < 0.01$; $^{****}P < 0.0001$ by ANOVA with the Dunnett's test for multiple comparisons.
(PDF)

**S5 Fig.** (A) Effects of antibodies against kininogen and vitronectin on the number of *C. glabrata* cells that were cell-associated with human endothelial cells. (B) Cell-association of the indicated number of C. glabrata cells that had been coated with either BSA or kininogen and vitronectin. (C) Effects of coating *C. glabrata* with BSA, kininogen, and/or vitronectin on the number of cell-associated organisms. (D) Flow cytometric detection of the binding of kininogen (top row) and vitronectin (bottom row) to *C. albicans* cells that had been incubated for 1 h with BSA alone, or with kininogen and vitronectin. Numbers in the upper right-hand corner indicate the percentage of positive cells. Results are representative of 4 separate experiments,

each of which analyzed 10,000 cells. (E-F) Summary of combined flow cytometry results showing the binding of kininogen (E) and vitronectin (D) to *C. albicans* cells. Data in (A-C) are the mean ± SD of 3 experiments each performed in triplicate. Orgs/HPF, organisms per high power field; ns, not significant; $^{***}P < 0.001$; $^{****}P < 0.0001$ by ANOVA with the Dunnett's test for multiple comparisons (A-C) or the Student's t-test (E and F).
(PDF)

**S6 Fig. Kininogen and vitronectin interact with gC1qR and αv integrins to induce adherence to human endothelial cells.** (A and B) Effects of BSA or kininogen and vitronectin on the endocytosis of the indicated strains of *C. albicans*. (C) Kininogen and vitronectin increase cell-association (adherence) of the indicated *Candida spp*. (D) Endothelial cell damage caused by the cells of the indicated strains that had been coated with kininogen and vitronectin. (E) Kininogen and vitronectin do not enhance the cell-association of *S. cerevisiae*. (F and G) Inhibition of cell-association of *C. glabrata* coated with either vitronectin alone (F) or vitronectin and kininogen (G) by antibodies against gC1qR (clone 74.5.2) and/or integrins αvβ3 and αvβ5. Data are the mean ± SD of 3 experiments each performed in triplicate. Orgs/HPF, organisms per high power field; ns, not significant; $^{*}P < 0.05$, $^{**}P < 0.01$, $^{***}P < 0.001$, $^{****}P < 0.0001$ by Student's t-test (A) or ANOVA with the Dunnett's test for multiple comparisons (B-G).
(PDF)

**S7 Fig. Mouse endothelial cells poorly support bridging molecule-mediated adherence.** (A and B) Cell-association of *C. glabrata* coated with either human or mouse serum by the indicated endothelial cells after 45 min (A) and 180 min (B). (C-F) Endocytosis (C and E) and cell-association (D and F) of *C. glabrata* coated with fresh human serum (C and D) or kininogen and vitronectin (E and F) by the indicated endothelial cells. (G) Western blot showing the protein levels of gC1qR, integrin αv and integrin β5 in mouse liver endothelial cells transduced with lentivirus containing the indicated human genes. gC1qR was detected with monoclonal antibody 60.1l, which only binds to the human protein. The integrins were detected with antibodies that recognize both human and mouse proteins. (H) Cell-association of *C. glabrata* coated with fresh human serum by mouse liver endothelial cells expressing human gC1qR, integrin αv, or integrin β5. Data in (A-F and H) are the mean ± SD of 3 experiments each performed in triplicate. HUVEC, human umbilical vein endothelial cell; orgs/HPF, organisms per high power field; ns, not significant; TIME, Tert-immortalized microvascular endothelial cells; $^{*}P < 0.05$, $^{**}P < 0.01$, $^{***}P < 0.001$, $^{****}P < 0.0001$ by ANOVA with the Dunnett's test for multiple comparisons (A, B and H) or the Student's t-test (C-F).
(PDF)

**S1 Table. Fungal strains used in this work.**
(PDF)

**S2 Table. Mean and standard deviation of the raw data from each endocytosis experiment.**
(XLSX)

## Acknowledgments

We thank Adam Diab for assistance with tissue culture and Nathan Wiederhold for generously providing the clinical fungal isolates.

## Author Contributions

**Conceptualization:** Quynh T. Phan, Scott G. Filler.

**Data curation:** Quynh T. Phan.

**Formal analysis:** Quynh T. Phan, Scott G. Filler.

**Funding acquisition:** Aaron P. Mitchell, Scott G. Filler.

**Investigation:** Quynh T. Phan, Norma V. Solis, Jianfeng Lin, Marc Swidergall, Hong Liu, Donald C. Sheppard.

**Methodology:** Jianfeng Lin, Marc Swidergall, Shakti Singh, Hong Liu, Ashraf S. Ibrahim, Aaron P. Mitchell.

**Project administration:** Scott G. Filler.

**Writing – original draft:** Quynh T. Phan, Scott G. Filler.

**Writing – review & editing:** Donald C. Sheppard, Ashraf S. Ibrahim, Aaron P. Mitchell, Scott G. Filler.

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
