## [Decision Letter · Decision Letter 0]

26 May 2022

Dear Scott,

Thank you very much for submitting your manuscript "Serum Bridging Molecules Drive Candida Invasion of Human but Not Mouse Endothelial Cells" for consideration at PLOS Pathogens. It was well received. As with all papers reviewed by the journal, your manuscript was reviewed by members of the editorial board and by independent reviewers. Reviewers appreciated the attention to an important topic. Based on the reviews, we are likely to accept this manuscript for publication, providing that you modify the manuscript according to the review recommendations.

Each of the reviewers asked for minor revisions.  Reviewer 1 also suggested some important textual modifications that acknowledge the limitations of the study.  Please seriously consider these recommendations in your revision.

Sincerely,

Bruce

Bruce S Klein

Associate Editor

PLOS Pathogens

Alex Andrianopoulos

Section Editor

PLOS Pathogens

Kasturi Haldar

Editor-in-Chief

PLOS Pathogens

orcid.org/0000-0001-5065-158X

Michael Malim

Editor-in-Chief

PLOS Pathogens

orcid.org/0000-0002-7699-2064

Reviewer Comments (if any, and for reference):

Reviewer's Responses to Questions

**Part I - Summary**

Reviewer #1: Fungal pathogens frequently grow in different morphotypes which can interact differently with host immune or non-immune cells. Here, the authors explored an important question about how C. albicans yeast cells interact with endothelial cells. They discovered that human serum proteins kininogen and vitronectin can adsorb to the yeast cell surface and enhance interaction of C. albicans and C. glabrata yeast with endothelial cells in culture. The receptors gC1qR and integrin cluster around adherent and endocytosed fungi, further implicating them actively in the process. Perhaps most interestingly, mouse serum proteins are not able to enhance interaction with mouse endothelial cells, which may help to explain why there are host-specific differences in the virulence of C. glabrata. These findings add significantly to our understanding of how C. albicans and C. glabrata may be able to interact with the host when it is in its yeast form. However, it is not clear whether these in vitro interactions are relevant in the context of infection, when there are many pattern recognition receptors and other opsonic receptors to mediate interactions with the host, and when the fungus is growing in the host rather than in a broth of YPD. In the absence of any experiments in mammalian infections, it is difficult to draw conclusions about the pathogenic impact of this mechanism.

Reviewer #2: This is a well-written and interesting manuscript examining endocyctosis of Candida by endothelial cells. The authors use multiple models and studies to show that several non-albicans Candida spp. bind vitronectin and kininogen bridging molecules. The interaction is more pronounce for human cells and serum compared to mice. The results are supported by the studies and the studies are easy to follow. The findings may help describe by some of the species are less virulent in mice when compared to C. albicans.

Reviewer #3: This manuscript describes a series of experiments that highlight the mechanisms that lead to endocytosis of non-albicans Candida species by endothelial cells, a necessary step in dissemination of Candida beyond the blood stream. Through a series of carefully conducted and clearly described experiments, specific serum components are identified that that act as bridging molecules to facilitate interaction with endothelial cell receptors that lead to endocytosis. Importantly, the manuscript also provides compelling evidence that these mechanisms are specific to human endothelial cells and deficient in mouse endothelium. This is a novel and significant finding in that murine models are in common use for disseminated candidiasis and these results offer a possible explanation as to why non-albicans yeast are notoriously well tolerated by mice in these models relative to C. albicans. Overall this is a well-executed study that provides important new mechanistic insights into these host-pathogen interactions with relevance to human disease.

**Part II – Major Issues: Key Experiments Required for Acceptance**

Reviewer #1: 1) Relevance of the findings to disease is not appropriately qualified. The host environment during infection is complex, with many cell types on the host side and different nutritional environments for the fungus. The impact of this work would be enhanced by some implication of these mechanisms during infection. One potential method would be to express human receptors in mice and add human serum proteins, to test the idea that increased interaction with endothelial cells through bridging serum proteins affects virulence of C. glabrata. This issue can also be addressed by appropriately qualifying the results to indicate that interactions in the whole mouse or human could be quite different from interactions in culture (e.g. add “mouse endothelial cells in culture” to line 338; edit lines 348/349 to read “mouse and human endothelial cells in culture”; add “in vitro” to the end of the sentence in line 355 and include the suggestion that these processes may be more complex in vivo).

Reviewer #2: None

Reviewer #3: None identified.

**Part III – Minor Issues: Editorial and Data Presentation Modifications**

Reviewer #1: 2) There can be quite a bit of variation among strains in a given species, including C. albicans (Marakalala et al. 2013) and C. glabrata (Ost et al 2021). This should be more fully discussed in the Discussion, given that it appears that only one representative of each strain was assayed.

Reviewer #2: Figure 1 Statistics: Verify the statistical analysis for 1 H and 1 I. Multiple comparison testing is listed, but it appears to only need single analysis.

Figures 1 and 2 legend: Clarify that these are human endothelial cells.

Figure 2 legend: Add statistical analysis description. For the antibodies, describe their different binding sites in the legend.

Line 109: Add a brief description about the relevance of the different C- and N-terminus antibody results.

Line 115: Were any negative results found (integrins that didn’t bind)? Those could be included in supplementary data.

Figure 3, 4, and 5 legend: Add that that the cells are human and add statistical analyses tests

Figure 5 D and E: Describe colors in legend or add to figure

Figure 7: Add statistical tests

Supplementary figures: Add statistical tests used and that the endothelial cells were human where relevant

Figure S6D and E: add a description of the colors for the bars

Reviewer #3: 1. The authors mention that "Although...yeast-phase C. parapsilosis cells are endocytosed by endothelial cells in vitro, this process is much slower and less efficient than the endocytosis of hyphal-phase C. albicans (Shintaku et al.). A limitation of these previous experiments is that they were performed in serum-free media." This sentence does not accurately reflect the data presented in the Shitaku paper. In fact, C. parapsilosis yeast underwent endocytosis with considerably higher efficiency than C. albicans hyphae (Fig. 3c) and in agreement with the present work, serum was required (Fig. 3d). The authors should reconsider these prior results in the context/discussion of their experiments.

2. The confocal micrographs in Fig. 3I are important data to support the role of gC1qR and avB5 as receptors for serum coated C. glabrata. Because the images with heat-inactivated serum are important to support the overall conclusions, it would be helpful if they were included in the same figure to facilitate side-by-side comparisons rather than in a supplemental figure.

3. Line 256 - The Western blot is actually Fig. S7G (not S7E)

4. Line 260 - These data are depicted in Fig. S7H (not S7G)

PLOS authors have the option to publish the peer review history of their article (what does this mean?). If published, this will include your full peer review and any attached files.

Reviewer #1: No

Reviewer #2: No

Reviewer #3: No

Figure Files:

Data Requirements:

Reproducibility:

References:

---

## [Editor Report · Decision Letter 1]

15 Jun 2022

Dear Scott,

We are pleased to inform you that your manuscript 'Serum Bridging Molecules Drive Candida Invasion of Human but Not Mouse Endothelial Cells' has been provisionally accepted for publication in PLOS Pathogens.

Best regards,

Bruce

Bruce S Klein

Associate Editor

PLOS Pathogens

Alex Andrianopoulos

Section Editor

PLOS Pathogens

Kasturi Haldar

Editor-in-Chief

PLOS Pathogens

orcid.org/0000-0001-5065-158X

Michael Malim

Editor-in-Chief

PLOS Pathogens

orcid.org/0000-0002-7699-2064
---

## [Editor Report · Acceptance letter]

4 Jul 2022

Dear Dr. Filler,

We are delighted to inform you that your manuscript, "Serum Bridging Molecules Drive Candidal Invasion of Human but Not Mouse Endothelial Cells," has been formally accepted for publication in PLOS Pathogens.

Best regards,

Kasturi Haldar

Editor-in-Chief

PLOS Pathogens

orcid.org/0000-0001-5065-158X

Michael Malim

Editor-in-Chief

PLOS Pathogens

orcid.org/0000-0002-7699-2064